# Aquathermolysis of Heavy Crude Oil: Comparison Study of the Performance of Ni(CH₃COO)₂ and Zn(CH₃COO)₂ Water-Soluble Catalysts

Yasser I. I. Abdelsalam [1], Firdavs A. Aliev [1,*], Oybek O. Mirzayev [1], Sergey A. Sitnov [1,*], Vladimir E. Katnov [1], Leysan A. Akhmetzyanova [2], Rezeda E. Mukhamatdinova [1] and Alexey V. Vakhin [1]

[1]   Institute of Geology and Petroleum Technologies, Kazan Federal University, 18 Kremlyovskaya Str., Kazan 420008, Russia; sailor-ya@mail.ru (Y.I.I.A.); mr.mirzayev92@mail.ru (O.O.M.); vkatnov@yandex.ru (V.E.K.); lkm.rm@mail.ru (R.E.M.); vahin-a_v@mail.ru (A.V.V.)
[2]   VNIIM-VNIIR, 7a 2nd Azinskaya Str., Kazan 420088, Russia; leisanvniir@yandex.ru
*   Correspondence: aquathermolysis@gmail.com (F.A.A.); sers11@mail.ru (S.A.S.)

**Abstract:** Aquathermolysis is one of the crucial processes being considered to successfully upgrade and irreversibly reduce the high viscosity of heavy crude oil during steam enhanced oil recovery technologies. The aquathermolysis of heavy oil can be promoted by transition metal-based catalysts. In this study, the catalytic performance of two water-soluble catalysts Ni(CH₃COO)₂ and Zn(CH₃COO)₂ on the aquathermolytic upgrading of heavy oil at 300 °C for 24 h was investigated in a high pressure–high temperature (HP-HT) batch reactor. The comparison study showed that nickel acetate is more effective than zinc acetate in terms of viscosity reduction at 20 °C (58% versus 48%). The viscosity alteration can be mainly explained by the changes in the group composition, where the content of resins and asphaltenes in the upgraded heavy crude oil sample in the presence of nickel catalyst was reduced by 44% and 13%, respectively. Moreover, the nickel acetate-assisted aquathermolysis of heavy oil contributed to the increase in the yield of gasoline and diesel oil fractions by 33% and 29%, respectively. The activity of the compared metal acetates in hydrogenation of the crude oil was judged by the results of the atomic H/C ratio. The atomic H/C ratio of crude oil upgraded in the presence of Ni(CH₃COO)₂ was significantly increased from 1.52 to 2.02. In addition, the catalyst contributed to the desulfurization of crude oil, reducing the content of sulfur in crude oil from 5.55 wt% to 4.51 wt% The destructive hydrogenation of resins and asphaltenes was supported by the results of gas chromatography-mass spectroscopy (GC-MS) and Fourier-transform infrared (FT-IR) spectroscopy analysis methods. The obtained experimental results showed that using water-soluble catalysts is effective in promoting the aquathermolytic reactions of heavy oil and has a great potential for industrial-scale applications.

**Keywords:** water-soluble catalysts; heavy oil; resins and asphaltenes; metal acetates; hydrothermal upgrading; viscosity reduction; nickel oxide

## 1. Introduction

In order to meet the increasing global demand for energy, which is expected to rise by more than 30% in the coming years, it is essential to create innovative methods for extracting the abundant reserves of unconventional hydrocarbons such as heavy oil, extra heavy oil, tar sands, oil shale, and bitumen [1]. Oil companies are showing interest in unconventional hydrocarbon resources as an alternative source of energy. These resources typically consist of highly viscous oils and their extraction requires additional efforts to ensure the efficiency of oil recovery from the reservoir. Even after the production of heavy oil and natural bitumen, their further transportation and refinery is challenging with the existing conventional methods and processes. Therefore, unconventional hydrocarbons need to be upgraded whether in situ or ex situ, while the former approach is more favorable.

In situ upgrading of heavy oil is primarily aimed to reduce the significant content of resins and asphaltenes, which are responsible for the high viscosity and many other problems associated with the refining of heavy oil. Resins and asphaltenes are high-molecular-weight compounds that are mainly composed of polycyclic aromatic hydrocarbons (PAHs), heteroatom-containing compounds and a wide variety of metals such as nickel, vanadium, iron, etc. [2]. These metals and heteroatoms cause a number of issues during the oil refining processes such as catalyst deactivation, corrosion, and equipment fouling. Therefore, it is rational to remove the heteroatoms and to reduce the content of high-molecular-weight fragments in the reservoir formations, which can be considered as the cheapest reactor ever gifted by nature. In situ upgrading contributes to the density and viscosity reduction by increasing the yield of the light distillate fractions such as gasoline and diesel. It can also help to reduce the environmental impact of heavy oil production, by minimizing the need for large-scale surface mining or other disruptive extraction methods. By upgrading the oil in-place, it is possible to reduce the amount of greenhouse gas emissions and other pollutants associated with heavy oil production.

There are several industrial technologies that are being developed for in situ upgrading of heavy oil and natural bitumen such as steam-assisted gravity drainage (SAGD), toe-to-heel air injection (THAI), electromagnetic heating, etc. [3]. Aquathermolysis is emerging in in situ upgrading technology, the term of which was derived from the Latin "aqua" = water, "thermos" = hot, and "lysis" = loosening, dissolution by Hyne et al. in early 1982 meaning the breaking down of the heavy fragments of crude oil through chemical reactions brought about by contact with steam [4]. The chemical reactions that make up the overall aquathermolytic process are numerous and are interrelated in a complex manner, which can be promoted by the catalysts. The importance of the catalytic role played by various metal ions in the various aquathermolytic reactions has been reported by many authors [5–8]. The conventional catalysts applied in the petroleum refining have been reported to be ineffective for the upgrading of unconventional hydrocarbons due to the significant content of resins, asphaltenes, various metals, and sulfur which poison the catalysts and deactivate them [9]. Moreover, injection of such catalysts into the reservoir formation is challenging and cost effective. Therefore, the development of fundamentally new catalysts is required due to the increasing production of unconventional hydrocarbons. The application of nanocatalysts in upgrading of heavy oil is expected to have a major impact on the petroleum industry. The advantages of nanocatalysts over traditional catalysts are increased surface area, more active sites, improved selectivity, lower operating temperatures, greater stability and resistance to deactivation [10].

Catalysts based on minerals, transition metal ions, their salts, and derivative complexes such as those used by Clark et al. [11], Chao et al. [12] were very active, providing viscosity reductions of up to 80% with significant yield of saturated and lighter aromatic compounds. However, their hot solubility in water means that catalyst recovery and subsequent reuse is fundamentally difficult. This can significantly limit the industrial value due to cost and disposal implications. Similarly, transition metal ions with an affinity for water, such as $Cu^{2+}$, $Fe^{3+}$, $Mo^{6+}$, and others, tend to form stable aqua complexes with water in the reaction medium, thereby limiting their availability to resins and asphaltenes in heavy oil. This factor is bound to reduce their viscosity reduction efficiency. Shokrlou and Babadagli [13] reported an improvement in viscosity reduction with increasing concentration and size of nanoparticles for Fe and Cu oxides. This also depended on the type of oxide. CuO provided the highest viscosity reduction under similar conditions. In all cases, the activity was also related to the degree of interaction between the catalyst particles and the resins or asphaltenes, the reaction temperature, and the degree of bond breaking. In a related approach using nanomaterials, Hendaningrath et al. [14] used nanoparticles of copper, zinc, nickel, and iron metals as catalysts. Their study showed that nanoscale metals are well suited for aquathermolysis due to a significant reduction in viscosity. The mechanism of action includes heat transfer, associated pyrolysis, and cleavage of catalytic bonds. Yang [15] showed that the naphthenate nickel catalyst can successfully crack most of the

asphaltene derivatives in heavy oil at 300 °C within 24 h. Yang et al. [16] developed an ultrasonic reactor for residue upgrading using tetralin as a hydrogen donor. Without tetralin, the viscosity decreased by 10.98%. However, when tetralin was added to the reactor, the viscosity was further reduced by 39.27%. While some recent studies [17,18] have suggested that these catalysts can be used under reservoir conditions to achieve high-quality crude oil, oil-soluble aquathermolysis catalysts face several challenges that limit their effectiveness in in situ upgrading processes. One of the main challenges is the difficulty of ensuring that the catalysts remain stable and active over extended periods of time in the harsh conditions present in reservoirs. The high temperatures, pressures, and corrosive environments can cause the catalysts to degrade or lose their activity, reducing their effectiveness. Moreover, the significant content of resins and asphaltenes under the thermal processes tends to result in the formation of coke-like substances due to the polymerization reactions. This carbon-rich solid material is an unwanted by-product that can accumulate on the surface of the catalysts, reducing their activity and selectivity over time. Another challenge of oil-soluble catalysts is the limited solubility in oil bulk. This can lead to poor distribution of the catalysts throughout the reservoir, resulting in uneven upgrading and reduced yields. In addition, the catalysts can also interact with other components in the oil, leading to unwanted side reactions and reduced selectivity. Finally, the cost of developing and deploying oil-soluble aquathermolysis catalysts can be a significant barrier to their widespread adoption. Despite these challenges, researchers and companies continue to explore the use of oil-soluble aquathermolysis catalysts as a means of improving the efficiency and sustainability of heavy oil production.

In contrast, water-soluble catalysts are less expensive to develop and to inject into the reservoir formations. They can be more easily and evenly distributed throughout the reservoir, leading to more consistent upgrading degrees. In addition, water-soluble catalysts are less likely to cause coking on the catalyst surface, as they can be more easily removed from the reservoir during the upgrading processes. Moreover, water-soluble catalysts are considered a clean and environmentally friendly material as they are easy to handle and transport in contrast to the oil-soluble catalysts, which are flammable and hazardous. Maity et al. [17] conducted steam upgrading experiments at 375–415 °C using water-soluble transition metals such as ruthenium (Ru) and iron (Fe) catalysts, resulting in a hydrodesulfurization effect of 21% and 18%, respectively. The authors also found improved formation wettability with the presence of $Fe^{2+}$ ions at higher temperatures (315–415 °C). Furthermore, although the asphaltene content increased from 14.6% to 19.7% by weight after the reaction, the viscosity of the crude oil decreased from 2140 to 520 mPa.s. In contrast, trivalent Ru (III) did not show this property. As a result, the authors concluded that the first series of transition metals and $Al^{3+}$ ions have high catalytic activity for thiophene and tetrahydrothiophene. These metal ions convert high-molecular-weight macromolecules into small molecules by breaking the C–S bonds. Zhong and Liu [18] investigated the catalytic effects of metal ions, including $Fe^{2+}$, $Fe^{3+}$, $Co^{2+}$, $Ni^{2+}$, $Cu^{2+}$, $Zn^{2+}$, $Mo^{2+}$, $Al^{3+}$, and $Mn^{2+}$, in the upgrading of heavy Liaohe oil under hydrothermal cracking conditions. The results showed that these metal ions contributed to the viscosity reduction of the heavy oil. In descending order, the activity of the metal ions was reported as follows: $Mo^{2+} > Co^{2+} > Fe^{2+} > Ni^{2+} > Al^{3+} > Cu^{2+} > Zn^{2+} > Mn^{2+}$. In particular, the viscosity-reducing effect of metal ions gradually increases with increasing reaction time and increasing reaction temperature. Notably, $Fe^{2+}$ can be a good candidate for metal catalysts because it has relatively low cost and good viscosity-reducing effect; its viscosity-reducing effect can reach 60% at 240 °C for 72 h. Studies by Yi et al. [19]. also confirmed this conclusion. A 90% viscosity reduction was achieved by Rivas et al. [20] using nickel sulfate to catalyze Venezuelan heavy oil. In addition, Zhang et al. [21] studied the catalytic effect of a water-soluble transition metal complex using heavy oil from the Yemen oil fields, which reduced viscosity by more than 70% at 180 °C. Clark et al. [22] also performed some catalytic upgrading experiments at 280 °C for 28 days. They used tetrahydrothiophene (THT) and thiophene (TP) as model oils. The experiments showed that the percentage of

sulfur removal for THT and TP was 64.0% and 55.0%, respectively. In another series of experiments under similar conditions, sulfur removal was 46% [23]. This suggested that metal salts could undergo cation exchange with the reservoir rock to form active sites to stimulate a heavy oil combustion reaction. Scanning electron microscopy (SEM) analysis of the clay surface also confirmed that metal cations were transferred from the liquid phase to the surface of the clay matrix to form active sites during the reaction. The authors reported that the water-soluble metal catalyst reduced the activation energy and ignition temperature of the reaction and increased oxygen consumption, making the oxidation process more efficient. Shallcross et al. [24] found that water-soluble tin and iron salts promoted coke formation, whereas water-soluble copper, nickel, and calcium salts had no significant effect on coke formation. Hyne et al., [25] Clark and Hyne (1990a [26] found that the use of a water-soluble metal salt catalyst promoted aquathermolysis. Fan et al. [27] reported that water-soluble catalysts can significantly promote aquathermolysis of crude oil. Li et al. [28] showed that nanoparticles obtained by microemulsion have good catalytic performance for aquathermolysis of heavy oil when they are suspended in the system, and nickel is a typical hydrogenation catalyst that can transfer hydrogen from steam to extra heavy oil. Inorganic salts, such as $VOSO_4$, $CuSO_4$, $NiSO_4$, $FeCl_2$, $ZnCl_2$, $SnCl_4$, $MnCl_2$, etc. [29,30] which are soluble in water, can serve as catalysts for aquathermolysis of heavy oil. Tang et al. [31] suggested that transition metal salts in combination with alkali (KOH/NaOH) were able to reduce the viscosity of heavy crude oil by 75–93% [32]. It should be noted that the water-soluble inorganic catalysts are economically and technologically efficient for implementation in oil fields.

The mechanisms of the catalytic reactions and thermal decomposition of metal acetates has been studied and well documented in the literature [33–36]. Doremieux J.L. studied pyrolysis of nickel acetate at 320 °C to reveal the decomposition mechanisms [36]. According to the author, nickel acetate tetrahydrate loses water with concurrent partial conversion to the basic salt accompanied by the evolution of some acetic acid. On continued heating, the constituent anion broke down to yield $CO_2$ and CO, acetic anhydride, acetone, and a residue containing both nickel carbide and nickel metal. Mohamed et al. reported the results of the thermal analysis of nickel acetate up to 500 °C. The authors found that parent salt dehydration occurred near 80 °C giving the anhydrous nickel acetate, which on further heating produced basic nickel acetate. Decomposition took place above 250 °C with the formation of nickel carbide ($N_3C$), which, together with the basic acetate, decomposed to a mixture of NiO and nickel metal. De Jesus et al. also studied nickel acetate tetrahydrate thermal decomposition in three different surroundings—air, helium, and hydrogen—by integrated thermogravimetric, quadrupole mass spectrometry, and X-ray photoelectron spectroscopy techniques [35]. The authors also implied that it is possible to synthesize metal nanoparticles from the thermal decomposition of nickel carboxylates. The authors reported the dehydration temperature of the nickel acetate between 118–137 °C with production of acetic acid due to the surface hydrolysis of the constituent anion. The further decomposition of the basic nickel acetate at 340 °C led directly to the formation of either NiO or Ni, under treatment atmospheres of air and $H_2$, respectively, while in an inert atmosphere, an additional step suggested that NiO is intermediate to the final production of metallic Ni with an estimated 4–5% of carbonaceous residues. The authors did not detect carbides as intermediates during the thermal decomposition of nickel acetate by XPS analysis. Overall, water-soluble catalysts have several advantages that make them a promising option for in situ upgrading of heavy oil. While there are still challenges to be overcome in developing and deploying these catalysts, ongoing research and development efforts are focused on improving their effectiveness and efficiency in the field.

In this paper, we report the catalytic performance of $Ni(CH_3COO)_2$ and $Zn(CH_3COO)_2$ on aquathermolysis of heavy oil sample at 300 °C carried out for 24 h. Both metals are well-known catalysts applied in oil refineries to promote destructive hydrogenation reactions, hydrodesulfurization, and water gas shift reactions. However, a limited number of experiments was conducted on the in situ aquathermolytic upgrading of heavy oil in the

presence of water-soluble catalysts. Oil properties such as viscosity, group composition, elemental and fractional compositions, FT-IR spectral coefficients, and GC-MS of saturates and aromatics were used to evaluate the outcome of the catalytic upgrading.

## 2. Results and Discussions

One of the major ways to understand the chemistry of heavy oil aquathermolysis and viscosity reduction is to establish the changes in the group composition of crude oil after catalytic upgrading. The results of heavy oil fractionation by the SARA-analysis method before and after non-catalytic and catalytic aquathermolysis are summarized in Figure 1.

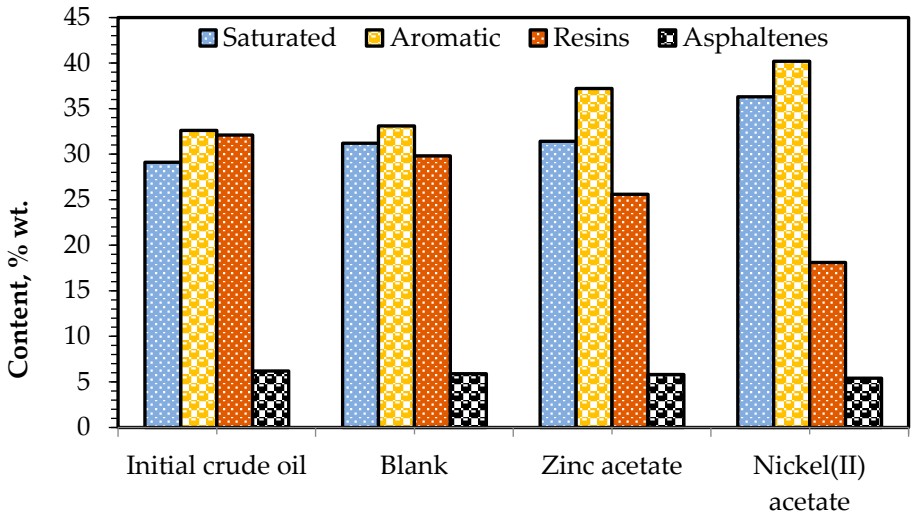

**Figure 1.** Group composition of heavy oil samples before and after aquathermolysis.

According to Hyne et al. the overall aquathermolytic reactions are triggered with the cleavage of S–S and C–S bonds, which can be relatively easy due to the chemical nature of the bonds compared to other bond types. Sulfur atoms have a relatively large atomic radius and the bond is relatively long which makes S–S bonds susceptible to cleavage even under mild thermal conditions (120 °C) [37–40]. Then, hydrocracked products can either react with steam to yield smaller fragments which can be involved in a further series of reactions such as alcohol rearrangement, decarbonylation, water–gas shift, hydrodesulfurization, etc., or polymerize in the absence (or lack) of hydrogen ions. These reactions are responsible for redistribution in the group composition of heavy oil samples. The content of resins and asphaltenes in the composition of the initial heavy crude oil sample was 32.1 wt% and 6.2 wt%, respectively. The aquathermolysis of heavy oil slightly reduced the content of resins (29.8 wt%). and asphaltenes (5.9 wt%). Zinc acetate contributed to the further reduction of resins (25.6 wt%), while the content of asphaltenes was almost the same with the content of blank sample asphaltenes (5.8 wt%) The destructive hydrogenation of resins in the presence of zinc increased the content of saturated and aromatics from 29.1 wt% and 32.6 wt% to 31.4 wt% and 37.2 wt%, respectively. The best catalytic performance in terms of improving the group composition quality of a heavy oil sample was shown by nickel acetate. It contributed to the reduction of resins by more than 43% in contrast to the initial sample, and 39% against the blank sample. At the same time, the destruction of asphaltenes is also significant, which indicates the high selectivity of nickel acetate in the destructive hydrogenation reactions. The content of asphaltenes was reduced from 6.2 wt% to 5.4 wt%, which corresponds to a reduction degree of almost 13%. The destructive hydrogenation products of asphaltenes increased the content of saturated and aromatic fractions by 25% and 23%, respectively.

The reason for the high viscosity of heavy oil samples is due to the significant content of high-molecular-weight compounds and their complex chemical compositions. Heteroatoms, such as sulfur, nitrogen, and metals, can have a significant influence on the

viscosity of heavy oil as well. These elements are often present in heavy oil as impurities, and they can contribute to the formation of highly condensed polycyclic structures that increase the viscosity of the oil. It is well-known that a small decrease in the content of the asphaltenes of oil can significantly reduce the viscosity of it. Even the changes in the molecular shape or conformation can alter the viscosity of heavy oil [41–43]. The results of viscosity measurements are illustrated in Figure 2. All upgraded oil samples were measured after degassing and separating the water phase from the oil phase by centrifuging at a temperature of 40 °C and at a rotation speed of 5000 rpm. The Ashalcha heavy oil corresponds to non-Newtonian fluids, the viscosity value of which is significantly reduced by the rise in temperature of the sample.

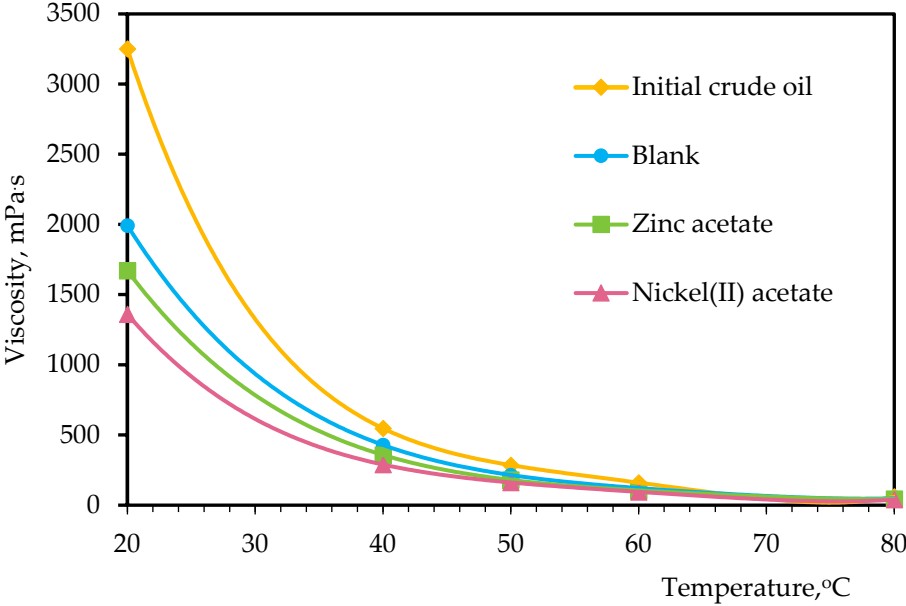

**Figure 2.** Viscosity of heavy oil samples before and after catalytic upgrading.

The viscosity of heavy oil samples correlates with the results of SARA-analysis. Non-catalytic aquathermolysis of heavy oil irreversibly reduced the viscosity of heavy oil sample from 3280 mPa.s to 1992 mPa.s measured at temperature of 20 °C due to the destruction of resins and asphaltenes. Zinc acetate contributed to the further viscosity reduction (1670 mPa.s), while the nickel acetate showed the best performance by reducing the viscosity of heavy oil sample down to 1359 mPa.s. This can be explained by the reduction of high-molecular-weight components such as resins and asphaltenes in the composition of the upgraded oil (see Figure 1). The nickel metal is probably more active than zinc in promoting the water–gas shift reaction, which is involved in a series of aquathermolysis reactions, to produce hydrogen ions. The protons in their turn will attack the alkyl substituted compounds formed after aquathermolytic cleavage of the organosulfur compounds found in the composition of resins and asphaltenes through hydrogenation reactions that will terminate the radicals from polymerization. Probably, the catalytic role of nickel is also greater in hydrogenation reactions. Thus, the content of light fragments of upgraded crude oil increase (see Figure 1) and the number of resins and asphaltenes significantly decrease, which is the reason for the highest viscosity reduction degree in the case of using nickel acetate—by 58%.

The elemental composition of heavy crude oil samples was studied to evaluate the upgrading process and quality of the catalytic aquathermolysis products. The results of elemental composition of heavy crude oil samples before and after non-catalytic and catalytic upgrading are presented in Table 1. The relative content of carbon decreases after non-catalytic and catalytic upgrading, while the share of hydrogen increases, justifying the hydrogenation of crude oil samples after aquathermolytic upgrading. The atomic H/C ratio

best describes the changes in the structure of the heavy oil samples after aquathermolytic upgrading. The initial heavy crude oil sample has a lower H/C ratio (1.52) than the upgraded crude oils (above 1.98), meaning that the initial oil sample is composed of larger, more complex hydrocarbon molecules that contain more carbon atoms per molecule. Generally, crude oil characterized with a low H/C ratio tends to yield more coke and less value-added products. The nickel acetate sample is characterized with the highest H/C ratio, which is in accordance with the results of the SARA-analysis method and viscosity measurements. In addition, both catalysts (zinc and nickel acetates) equally contribute to the reduction of sulfur content from 5.55 wt% to 4.5 wt% There are two possible reaction pathways for sulfur to be removed—$H_2S$ as an off gas during the steaming process and sulfidation of the corresponding metal oxides during the catalyst activation phase.

**Table 1.** Elemental composition of heavy oil samples before and after aquathermolysis.

| Samples | Contents, wt% | | | | |
|---|---|---|---|---|---|
| | C | H | N | S | $H/C_{atomic}$ |
| Initial oil sample | 83.34 | 10.63 | 0.48 | 5.55 | 1.52 |
| Blank | 81.80 | 13.52 | 0.00 | 4.68 | 1.98 |
| $Zn(CH_3COO)_2$ | 81.98 | 13.50 | 0.00 | 4.51 | 1.98 |
| $Ni(CH_3COO)_2$ | 81.65 | 13.71 | 0.00 | 4.54 | 2.02 |

Another oil quality diagnostic test is the atmospheric distillation of crude oil, the results of which allows any effects of the metal acetates on the fraction yield to be observed. The fractionation of the crude oil samples is possible due to the difference in the boiling point ranges. The fractions of heavy oil samples boiling up to 200 °C called light or gasoline fractions, from 200 to 300 °C medium or diesel fractions, and over 300 °C corresponding to the atmospheric distillation residue are presented in Table 2. The table shows the fraction yield in weight percent for each sample at different boiling point ranges, with the coke fraction also included. The results show that the $Ni(CH_3COO)_2$ catalyst contributed to the reduction of the initial boiling point from 168 °C to 120 °C, and also leads to an increase in gasoline and diesel fractions by 25% and 30%, respectively. The residue fraction within the boiling range of 300 °C and higher was reduced by 5.2 wt% in contrast to the residue fraction of the initial oil sample. The $Zn(CH_3COO)_2$ catalyst was less effective than $Ni(CH_3COO)_2$ on increasing the yield of light fractions and reducing the initial boiling point temperature (Table 2). The obtained changes in the atmospheric distillation fractions after nickel acetate-assisted aquathermolysis of heavy oil indicates the improvement in the physical and chemical properties of the oil, and its performance for upgrading and refining.

**Table 2.** Atmospheric distillation fractions before and after aquathermolysis.

| Samples | i.b.p., °C | Fraction Yield, wt% | | | Total, wt% |
|---|---|---|---|---|---|
| | | i.b.p.–200 °C | 200–300 °C | above 300 °C | |
| Initial oil sample | 168 | 1.5 | 13.9 | 84.6 | 100 |
| Blank | 156 | 1.7 | 15.8 | 82.5 | 100 |
| $Zn(CH_3COO)_2$ | 125 | 1.9 | 16.2 | 81.9 | 100 |
| $Ni(CH_3COO)_2$ | 120 | 2.0 | 17.9 | 80.1 | 100 |

FT-IR Spectroscopy is an analytical tool applied to evaluate the upgrading performance of the studied catalysts by providing information about the changes in the functional groups, which is considered as additional supportive data to the structure and chemical composition of heavy oil. Based on the resulting spectra (Figure 3), coefficients such as $C_1$-aliphaticity, $C_2$-aromaticity, $C_3$-branching, $C_4$-condensation, $C_5$-oxidation and $C_6$-, $C_7$-sulfurization degree were estimated according to [44].

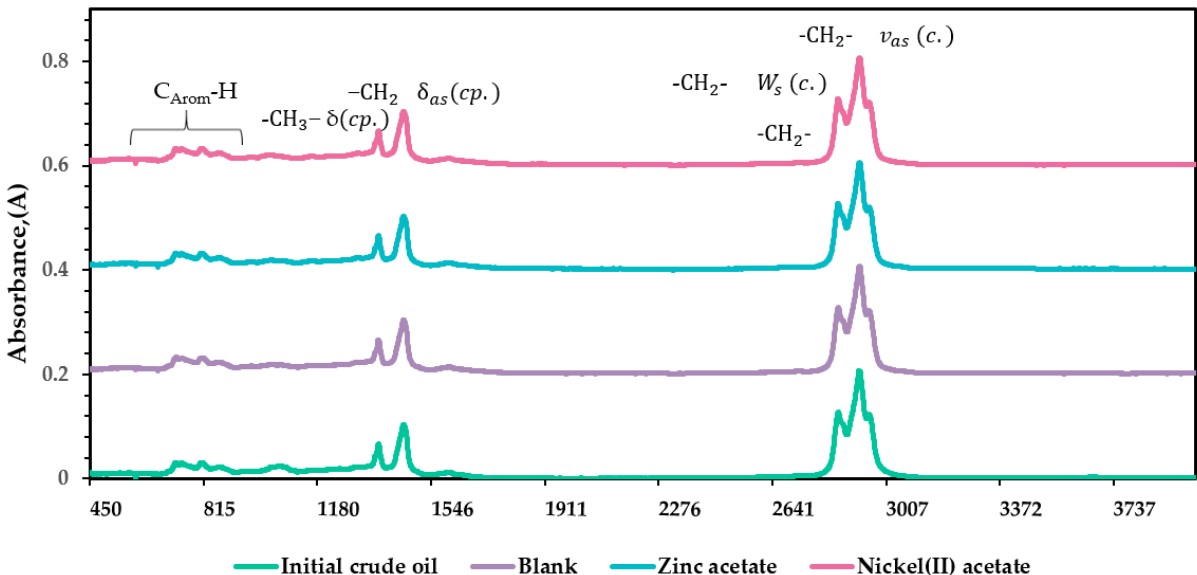

**Figure 3.** FTIR spectra of crude oil samples.

Spectral coefficients were calculated according to the following equations: $C_1 = D_{1450}/D_{1600}$, aliphaticity (shows the proportion of C–H bonds in aliphatic structures with respect to C=C bonds in aromatic structures); $C_2 = D_{1600}/D_{720+1380}$, aromaticity (shows the proportion of C=C bonds in aromatic groups with respect to C–H bonds in aliphatic structures); $C_3 = D_{1380}/D_{720}$, branching (shows the proportion of methylene groups to methyl groups); $C_4 = D_{1600}/D_{740+860}$ degree of condensation (shows the proportion of C=C bonds in aromatic groups with respect to C–H bonds in aromatic structures;); $C_5 = D_{1700}/D_{1600}$, degree of oxidation (shows the proportion of carbonyl groups R–C=O (in the presence of an OH-group) with respect to C=C bonds in aromatic structures); $C_6 = D_{1030}/D_{1600}$, degree of sulfurization (shows the proportion of S=O-bonds in sulfoxide groups (either in sulfonates or sulfonic acids provided that absorption bands are in the range of 1260–1150 cm$^{-1}$ and 700–600 cm$^{-1}$) with respect to C=C bonds in aromatic groups; sulfonates); $C_7 = D1_{160}/D_{1600}$, degree of sulfurization (shows the proportion of S=O-bonds in sulfonate groups with respect to C=C bonds in aromatic groups, sulfoxides).

From the above data we can conclude that there is a decrease in the values of the coefficients of aliphaticity, branching, and sulfur content with a simultaneous increase in condensation and oxidation, which clearly indicates the compaction of the naphthenic molecules and substitution of naphthene-alkyl fragments with sulfur- and oxygen-containing functional groups (Table 3).

**Table 3.** Spectral coefficients of crude oil before and after aquathermolysis.

|  | $C_1$ | $C_2$ | $C_3$ | $C_4$ | $C_5$ | $C_6$ | $C_7$ |
|---|---|---|---|---|---|---|---|
| Crude oil | 8.5 | 0.5 | 1.9 | 0.4 | 0.3 | 1.9 | 1.3 |
| Blank | 7.2 | 0.5 | 1.8 | 0.5 | 0.5 | 1.4 | 1.2 |
| Zn(CH$_3$COO)$_2$ | 7.4 | 0.5 | 1.8 | 0.5 | 0.5 | 1.4 | 1.2 |
| Ni(CH$_3$COO)$_2$ | 7.1 | 0.5 | 1.7 | 0.5 | 0.5 | 1.5 | 1.2 |

The GC-MS spectra of saturates before and after catalytic upgrading are provided in Figure 4.

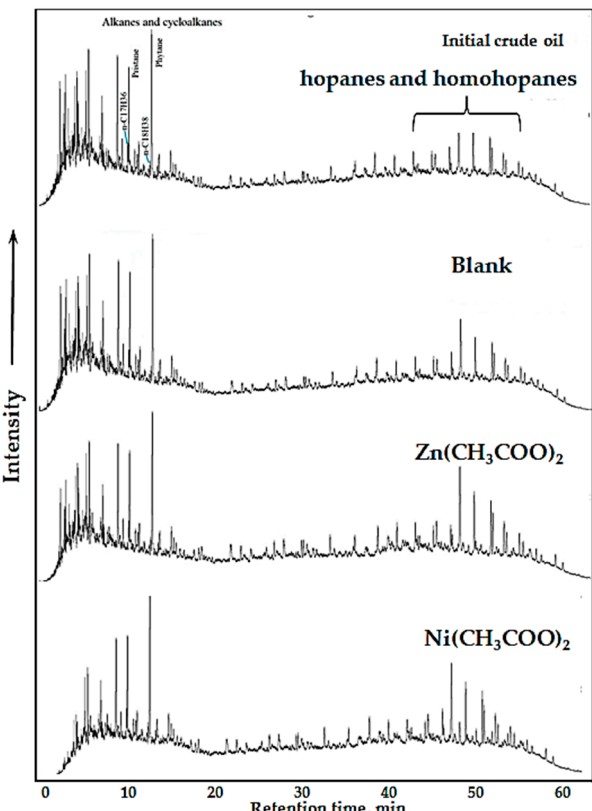

**Figure 4.** The GC-MS spectra (TIC) of saturates of crude oil samples.

The obvious changes were detected in the composition of hopanes and its homologues corresponding to a retention time of above 40 min. The intensity of *n*-alkanes, which correspond to the retention time below 10 min, changes after catalytic upgrading. The quantitative distribution of the low-molecular weight normal alkanes C10–C20 in saturates is presented in Figure 5.

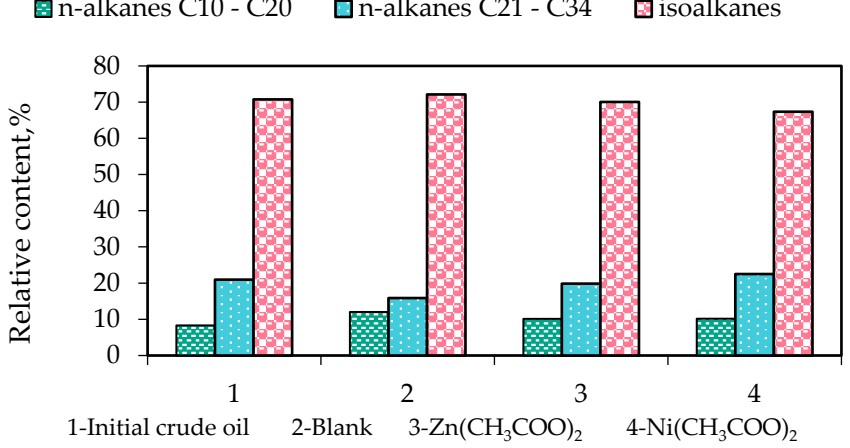

**Figure 5.** Relative content of *n*- and iso-alkanes in the composition of saturated hydrocarbons.

The relative content of normal C10–C20 alkanes in the composition of saturates after non-catalytic aquathermolysis was raised from 8.3% to 12%. Addition of the catalysts contributed to the reduction of the C10–C20 n-alkanes from 12% to 10%, while the content of high-molecular weight n-alkanes (C21–C34) was raised from 16% to 23%. The relative content of iso-alkanes was reduced from 72% to 67% in the case of nickel acetate. It can be assumed that as a result of aquathermolysis, radical addition reactions occur in the control sample, resulting in the formation of alkanes with a longer C10–C20 hydrocarbon

chain. The increase in the number of n-alkanes occurs as a result of degradation of resin and asphaltene molecules due to stripping of aliphatic substituents, as well as dealkylation reactions of cyclic saturated and aromatic hydrocarbons.

In Figure 6, the content of saturated hydrocarbons (alkanes) is compared with alkenes and hopanes. It was found that 61% of the saturated fraction of the initial crude oil sample was composed of alkanes, while cycloalkanes made only 16% of the fraction. The non-catalytic aquathermolysis of heavy oil contributed to the reduction of alkanes from 60% to 52% and increased the share of alkenes by almost twice. Probably, the intensive hydrogenation of alkenes in the presence of the catalysts increased the share of alkanes, such that in the case of Zn acetate the amount of alkenes was reduced from 30% to 17%. The catalytic hydrogenation products of alkenes probably increased the content of alkanes from 52% to 57%. The relative content of alkenes was further decreased (by almost twice) in the case of Ni acetate, which led to the increase in the content of alkanes from 52% to 60%.

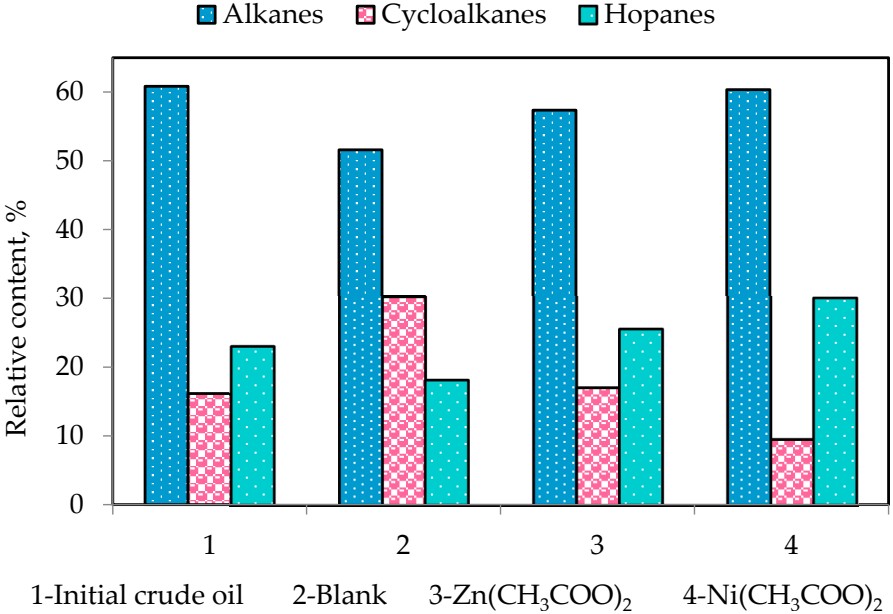

**Figure 6.** Relative content of alkanes, cycloalkanes, and hopanes in the composition of saturated hydrocarbons.

The GC-MS spectra of aromatics fractions as per total ion chromatogram (TIC) are presented in Figure 7. The reduction in the intensity of peaks corresponding to the retention time range of 10–35 min in the TIC of aromatics after the non-catalytic aquathermolysis process indicates the transformation of benzenes, naphthalenes, and thiophenes either due to the hydrogenation of aromatic hydrocarbons or the scission of alkyl radicals from benzene structures with further polymerization of the hydrocracked products. The aquathermolysis of heavy oil in the presence of both catalysts contributed to the significant increase in the intensity of peaks corresponding to 5,7-dimethyltetralin, alkyl benzenes $C_{14}H_{22}$, 2,5,7-trimethyl-1-benzothiophene, 4-methyl-dibenzothiophene in contrast to the spectrum of a blank sample. These compounds are of high-molecular-weight, which originally concentrate in the resin and asphaltene fractions. It means the products of the resin and asphaltene destructive reactions increased the content of the mentioned compounds in the aromatics, thus showing the performance of zinc and nickel acetate in destructive hydrogenation of resins and asphaltenes.

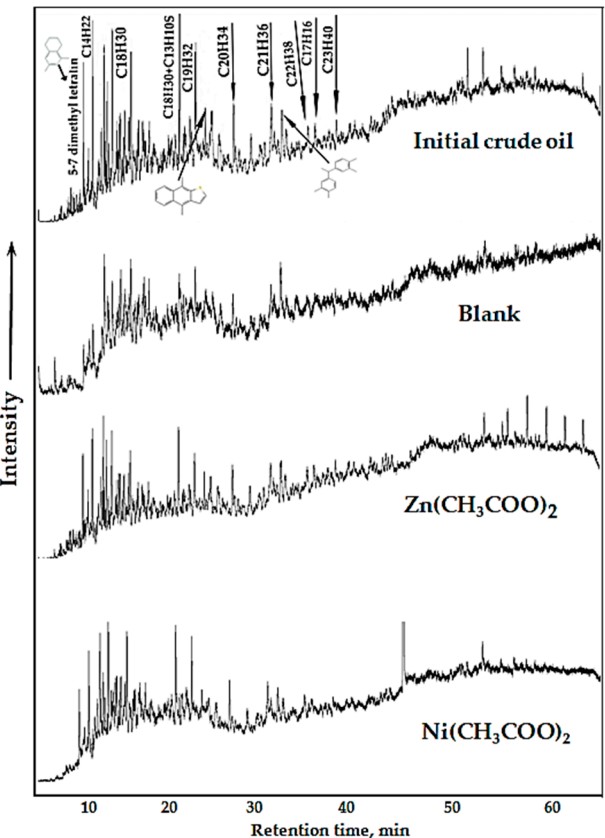

**Figure 7.** The GC–MS spectra (TIC) of aromatics of crude oil samples.

The quantitative analysis of the aromatic hydrocarbon compounds is illustrated in Figure 8. The relative content of alkylbenzenes was reduced from 29% to 24% after non-catalytic upgrading. The introduction of both catalysts equally increased the content of alkyl-benzenes up to 27%. The content of naphthalene in the aromatic fraction increased from 9% (blank) to 15% after catalytic upgrading. A slight increase in the content of biphenyls and other compounds was observed after the non-catalytic and catalytic processes.

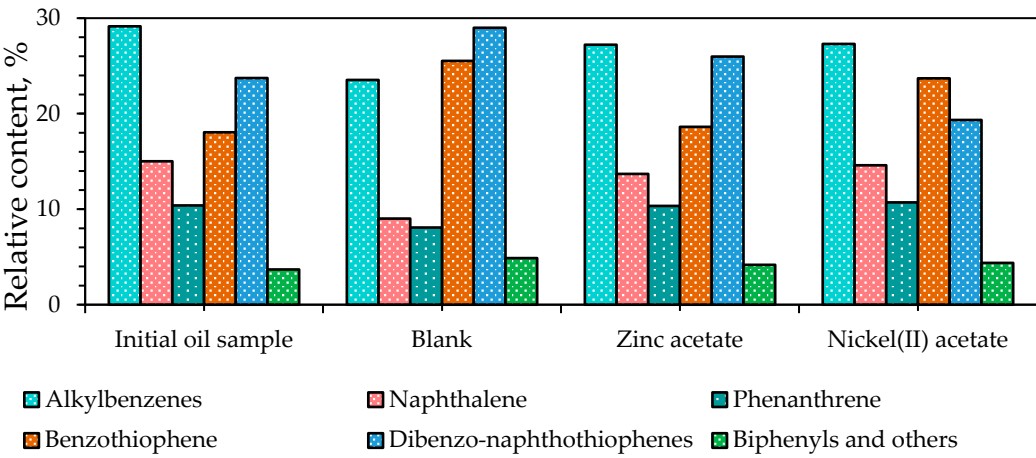

**Figure 8.** Relative content of aromatic compounds in the composition of aromatics fractions.

It was interesting to study the thermal decomposition products of metal acetates after the aquathermolysis processes and their distribution in the reaction mixture. For this purpose, water was separated from the oil phase by centrifuging at a rotation speed of 5000 rpm and at a temperature of 40 °C. The metal concentration was expected to be

observed in the water phase in the form of metal oxides (i.e., nickel (II) oxide) or just metal (i.e., nickel). However, no precipitates were observed in the bottom of the cylinders after centrifuging, which means either the metal acetate was not decomposed under the given thermobaric conditions, or the active forms of the catalysts were dispersed in the oil phase. To check this out we evaporated the water at a temperature of 60 °C and analyzed the solid residue by elemental analysis and energy-dispersive X-ray spectroscopy (EDX), the results of which are demonstrated in Table 4 and Figures 8 and 9. The results showed that the concentration of nickel in the composition of nickel acetate before the aquathermolysis reaction was 217,028 ppm and was reduced to 1351 ppm after the aquathermolysis reaction. The concentration of zinc in Zn acetate before the aquathermolysis reaction was 339,387 ppm and it was also reduced after the aquathermolysis reaction to 10 ppm. This means that a significant part of the thermal decomposition product of the metal acetates, which is probably in the form of the metal oxide, remained and dispersed in the oil phase.

**Table 4.** Results of elemental composition of the water residue.

| Samples | | Elements, ppm | |
| --- | --- | --- | --- |
| | | Ni | Zn |
| **Ni acetate** | Before aquathermolysis | 217,028.3588 | 0.2623 |
| | After aquathermolysis | 1350.9757 | 93.8973 |
| **Zn acetate** | Before aquathermolysis | 4.4831 | 339,386.7946 |
| | After aquathermolysis | 61.2142 | 10.3284 |

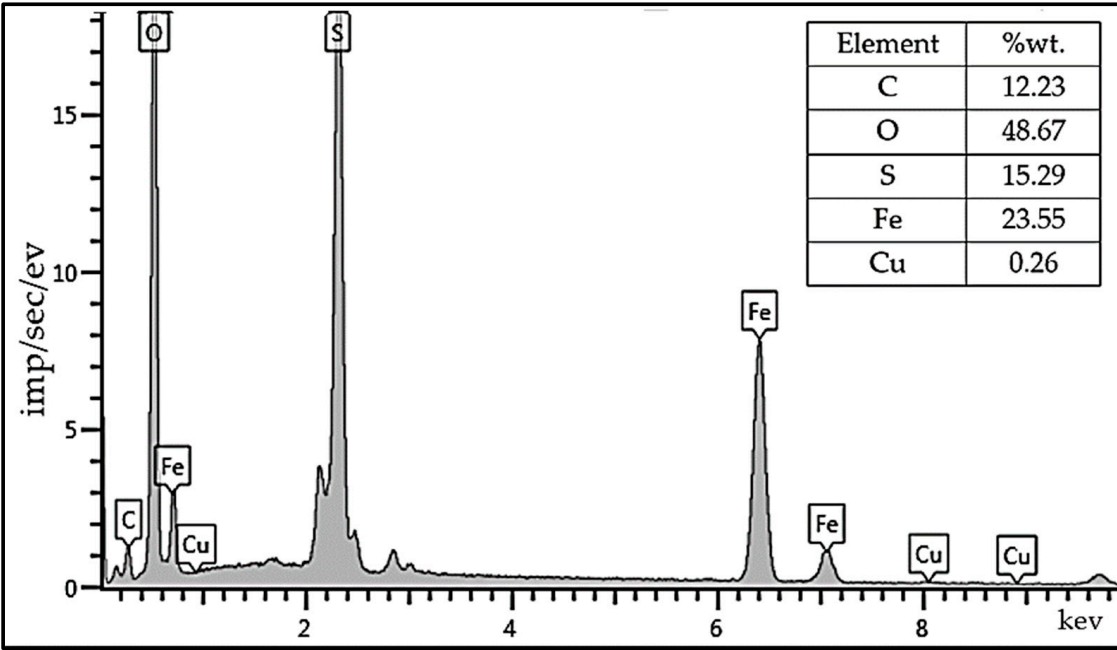

**Figure 9.** The EDX spectrum of the water residue after aquathermolysis in the presence of zinc acetate.

The results of EDX mapping of the water residue in the case of zinc acetate (Figure 9) and nickel acetate (Figure 10) revealed the absence of the nickel and zinc oxides in the water phase, which means the metal acetates decomposed under the aquathermolytic rection conditions and distributed in the oil phase as no metals were found in the water phase. Further studies with high concentration (above 10 wt%) of the catalysts, which take the distribution of the active form of the water-soluble catalyst in the oil phase into account, will need to be performed.

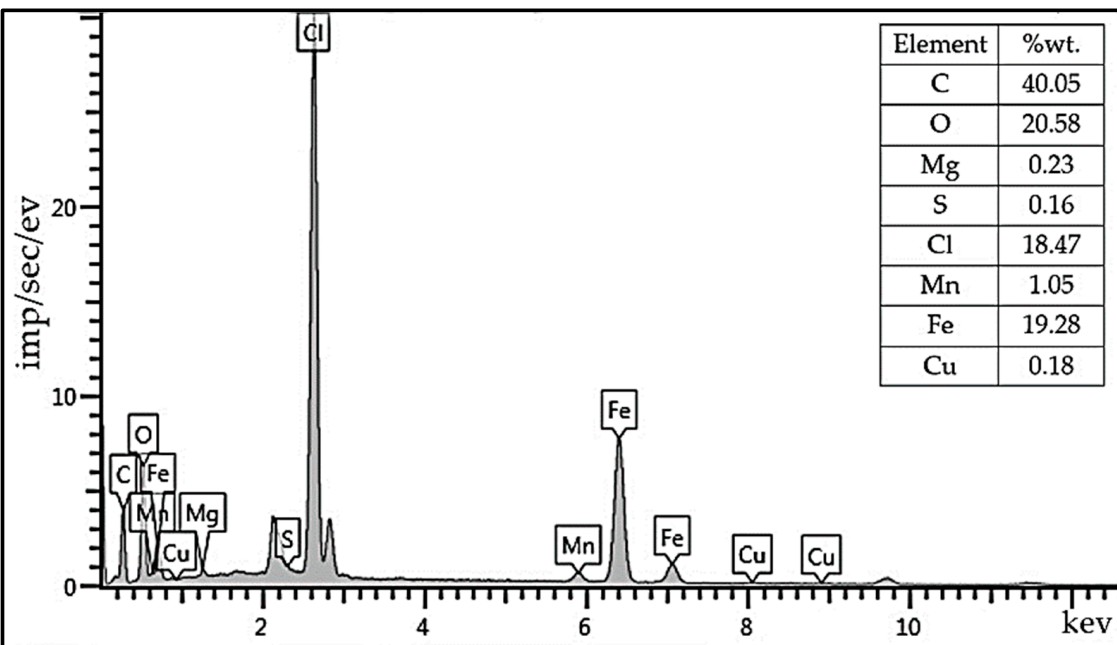

| Element | %wt. |
|---------|------|
| C | 40.05 |
| O | 20.58 |
| Mg | 0.23 |
| S | 0.16 |
| Cl | 18.47 |
| Mn | 1.05 |
| Fe | 19.28 |
| Cu | 0.18 |

**Figure 10.** The EDX spectrum of the water residue after aquathermolysis in the presence of Nickel acetate.

## 3. Experimental Part

### 3.1. Object of Research

In the present study, we used a heavy oil sample obtained from the Aschalcha field in the republic of Tatarstan (Russia). Aquathermolysis processes in the presence of water soluble catalysts $Ni(CH_3COO)_2$ and $Zn(CH_3COO)_2$ were carried out in a laboratory installation, consisting of a reactor, heating and mixing devices, as well as a control unit for monitoring the progress of the process and recording the kinetics of the process. The diagram of the autoclave is shown in Figure 11. The temperature was raised to 300 °C in less than 1 h and maintained at 300 °C for 24 h. The autoclave was then cooled to room temperature. The initial pressure was supplied by inert nitrogen gas equal to 10 bar.

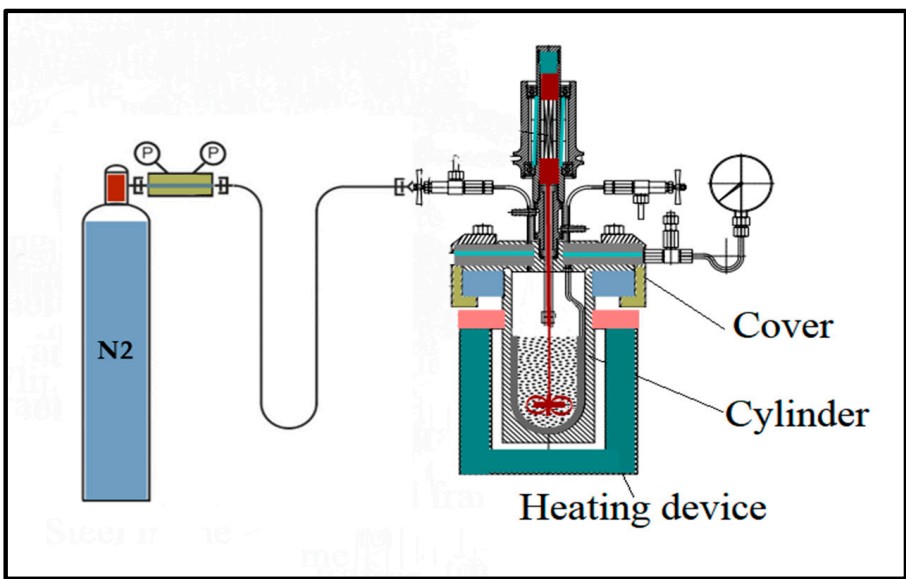

**Figure 11.** Schematic illustration of high-pressure reactor.

The model system was composed of 70 g of a heavy oil sample and 30 g of distilled water. The concentration of the metal acetate was chosen based on previous experiences and literature data—2 wt% in all experiments.

### 3.2. CHNS-Analysis of Upgraded Crude Oil

Analyzer Perkin Elmer 2400 Series II (Perkin Elmer, MA, USA) was used to analyze the elemental composition of the heavy oils before and after upgrading. It allows the determination of the content of carbon, hydrogen, nitrogen, oxygen, and sulfur in the oil.

### 3.3. FTIR of Crude Oil before and after Upgrading Experiments

FTIR spectroscopy was performed in a Vertex 70 FTIR spectrometer (Bruker, ET, Germany) in order to reveal the possible structural changes in the heavy oil before and after hydrothermal upgrading in the absence and presence of the developed catalyst.

### 3.4. Atmospheric Distillation of Crude Oil

The atmospheric distillation of the crude oil and conversion products after the autoclave reaction was carried out used an ARN-LAB-03 distillation plant, consisting of a heater and a condenser.

### 3.5. Viscosity Measurements

The viscosity of the oil samples before and after the aquathermolytic upgrading were measured using a FUNGILAB Alpha L rotational viscometer equipped with a thermostatically controlled jacket at a temperature range of 20–80 °C with steps of 10 °C. The required temperature in the jacket was maintained using a HUBER MPC K6 cooling thermostat. The volume of samples was 6.7 mL and a TL5 spindle was used. The shear rate for the given spindle was determined by multiplying the rotation speed by 1.32 (as per passport of the equipment). The relative error and repeatability of the viscosimeter are ±1.0% and 0.2%, respectively.

### 3.6. SARA-Analysis

The group composition of heavy oil samples before and after aquathermolytic upgrading was determined by separating them into four fractions: saturates, aromatics, resins, and asphaltenes according to the standards of ASTM D2007. The ratio of oil:hexane to precipitate asphaltenes from the oil bulk system was 1:40. Further, maltenes were separated into saturates, aromatics, and resins based on the solubility of each fraction in various solvents. Aluminum oxide calcined at 420 °C was used as an adsorbent.

### 3.7. Gas Chromatography-Mass Spectroscopy (GC-MS)

The Saturates and aromatic fractions obtained from the experiments were analyzed by a GC-MS system, which included the gas GC "Chromatech-Crystal 5000" with a mass-selective detector ISQ (Waltham, MA, USA). The obtained results were processed by Xcalibur application. First, a capillary column 30 m long and diameter 0.25 mm was used to perform the experiments. The carrier gas was helium with a flow rate of 1 mL/min at a temperature of 310 °C. The adjusted thermostat regime started with a rise from 100 °C to 150 °C and a heating rate of 3 °C/min, from 150 °C to 300 °C with a heating rate of 12 °C/min followed by an isotherm to the end of the analysis. Finally, the electron energy was 70 eV and the ion source temperature was set at 250 °C. Saturate and aromatic fraction elements were identified by means of the NIST Mass Spectral Library and literature sources.

## 4. Conclusions

This paper investigated the catalytic upgrading performance of two water-soluble catalysts on aquathermolysis of heavy oil samples from the Ashalcha reservoir. The experimental results showed that nickel acetate is more effective than zinc acetate in terms of viscosity reduction. The viscosity reduction degree of 58.6% was achieved with the

aquathermolysis of heavy oil carried out at 300 °C for 24 h. The good performance of nickel acetate to improve the rheology of heavy oil is attributed to the catalytic ability of $Ni(CH_3COO)_2$ to destroy the resins and asphaltenes and hydrogenate the destruction products, such that the content of resins was reduced from 32.1 wt% to 18.1 wt%, and asphaltenes from 6.2 wt% to 5.4 wt% Moreover, it was found that nickel acetate is also more active in hydrogenation, hydrodesulfurization, and hydrodenitrogenation of heavy crude oil during aquathermolysis. The atomic H/C ratio of heavy crude oil was increased from 1.52 to 2.02 and the content of sulfur was reduced from 5.55 wt% to 4.54 wt% The nitrogen element was totally removed after catalytic upgrading. In addition, nickel acetate-assisted aquathermolytic upgrading resulted in the reduction of the initial boiling point temperature from 168 °C to 120 °C, which is explained by the increase in the gasoline and diesel fractions of 33% and 29%, accordingly. FT-IR and GC-MS Spectroscopy analytical tools were applied to evaluate the upgrading performance of the studied catalysts by providing information on the changes in the composition of saturates and aromatics and the functional groups in the heavy crude oil. This was considered as additional supportive data in determining the structure and chemical composition of the heavy oil. The evidence from this study suggests that water-soluble catalysts are effective in promoting aquathermolytic reactions.

**Author Contributions:** Conceptualization, Y.I.I.A. and A.V.V.; methodology, V.E.K.; investigation, O.O.M. and Y.I.I.A.; resources, R.E.M. and S.A.S.; writing—original draft preparation, F.A.A.; writing—review and editing, F.A.A.; supervision, A.V.V.; funding acquisition, L.A.A.; data curation, F.A.A. and Y.I.I.A. All authors have read and agreed to the published version of the manuscript.

**Funding:** This work was supported by the Russian Science Foundation (Grant No. 22-73-00021).

**Data Availability Statement:** Not applicable.

**Conflicts of Interest:** The authors declare no conflict of interest.

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
