# Peer review of "Aquathermolysis of Heavy Crude Oil: Comparison Study of the Performance of Ni(CH3COO)2 and Zn(CH3COO)2 Water-Soluble Catalysts"

_catalysts, doi:10.3390/catal13050873_

Round 1

Reviewer 1 Report

The authors have presented a study regarding the in-situ upgrading of heavy oil through aquathermolysis by Ni and Zn based compounds for enhanced oil recovery. The quality of this manuscript is good. However, there are some places (listed below) need to be improved before this manuscript could be considered for publication in Catalysts.

(1)  The authors claimed that the Ni(CH3COO)2 and Zn(CH3COO)2 were water soluble. What was the water content during the aquathermolysis of each heavy oil at 300 centigrade for 24 h ? 

(2) Did the authors had investigated the effect of water content (or water/oil ratio) on the performances of Ni(CH3COO)2 and Zn(CH3COO)2 ?

(3) Table 2,  2.0% +  18.0% + 80.1% = 100% ?

(4) In this manuscript, the performance of Ni(CH3COO)2 was found to be higher than that of Zn(CH3COO)2, it was  better if some explanations could be provided.

The quality of this manuscript is good.

Reviewer 2 Report

The manuscript is well written and should be published.  The effects of the catalysts are not very strong, but in principle they reduce the viscosity to a statistically significant extent.

Author Response

Thank you very much for your positive feedback. We appreciate the time and effort you have dedicated to providing your valuable feedback on our manuscript.

Reviewer 3 Report

The present paper shows the investigation of the catalytic upgrading performance on aquathermolysis reaction of heavy oil sample with use of Ni acetate and Zn acetate as a catalyst. In spite of a few novel and original findings, it cannot be accepted for the publication, unless it will be majorly revised. See the following comments.

1.    We cannot find any discussion on the mechanistic investigation of this catalysis. It is the most important for physico-chemical journal. How did Ni acetate catalyze it? What was the role of Ni? Zn is the same.

2.    First of all, is this a catalytic reaction? Is Ni or Zn a reactant, not a catalyst? From this point of view, the authors should clarify the behavior of Ni and Zn in a whole reaction system. Namely, did authors check the recovery of Ni and Zn?

3.    If the decomposition of acetates of Ni and Zn occurred, where were (hydr)oxides of Ni and Zn located? The authors should show them with the direct evidences.

4.    Regarding the catalytic effect, are salts with ions other than acetate ions useless? Why did the authors choose acetate?

Round 2

Reviewer 3 Report

The reviewer understands and accept the author's response, and I look forward to the submission of a follow-up to this paper.